# Metabolome and Transcriptome Profiling Reveal That Four Terpenoid Hormones Dominate the Growth and Development of *Sanghuangporus* *baumii*

**DOI:** 10.3390/jof8070648

**Published:** 2022-06-21

**Authors:** Zengcai Liu, Xinyu Tong, Ruipeng Liu, Li Zou

**Affiliations:** College of Forestry, Northeast Forestry University, Harbin 150040, China; 1758458181@nefu.edu.cn (Z.L.); hhxxyx@nefu.edu.cn (X.T.); liuruipeng@nefu.edu.cn (R.L.)

**Keywords:** *Sanghuangporus* *baumii*, growth and development, metabolome and transcriptome, isopentenyl diphosphate isomerase, terpenoid biosynthesis, hormone

## Abstract

*Sanghuangporus* *baumii* is a traditional medicinal fungus that produces pharmacological terpenoids, but natural resources are insufficient for applications, and its growth and development mechanisms are poorly understood. Combining metabolomic and transcriptomic analyses, we found four terpenoid hormones and a central gene, isopentenyl diphosphate isomerase (*IDI*), involved in growth and development. Additionally, an exogenous hormone test was used to further confirm the importance of the four terpenoid hormones. Finally, hormone content determination and qRT−PCR were performed to explore the growth and development mechanism; we found thatcis-zeatin (CZ) plays a major role in the mycelia stage, trans-zeatin (TZ) and gibberellin A4 (GA4) are important in the primordia stage, GA4 is crucial for the fruiting bodies stage, and abscisic acid (ABA) may be a marker of maturity. The *IDI* gene was also found to affectterpenoid hormone content by regulating the relative gene transcript levels, thereby controlling morphological changes in *S. baumii*. Our results revealthe growth and development mechanisms of *S. baumii* and may promote the breeding and utilisation of high-quality varieties.

## 1. Introduction

*Sanghuangporus baumii* (Pilát) L.W. Zhou & Y.C. Dai is a precious fungus with medicinal effects, including improving immunity, a hypoglycemic effect and anticancer and antitumour activities [1,2]. The market demand for *S. baumii* is increasing, and this is gradually exhausting wild resources [3]. Wild-type (WT) *S. baumii* is host-specific and grows only on *Syringa reticulata*. It grows slowly, taking 1–2 years to develop mature fruiting bodies suitable for human use. Due to the scarcity of its *S. reticulata* host in the wild and the slow growth rate of *S. baumii*, existing *S. baumii* wild resources are being depleted; hence, domestication and cultivation are being actively explored.

The growth and development processes of *S. baumii* involve growth from mycelia (Myc) to primordia (Pri), followed by the formation of fruiting bodies (Fru). However, because the growth and development mechanism of *S. baumii* remains unclear (Figure 1), the yield is unstable, and the quality is uneven. As molecular biology technology has developed, metabolome and transcriptome studies have been applied to uncover growth mechanisms in greater detail [4,5]. Changes in metabolites and corresponding genes associated with growth and development can be measured [6]. Therefore, metabolome and transcriptome analyses could expand our understanding of changes in *S. baumii* at different developmental stages.

Terpenoidmetabolites are often important molecules in medicinal research, as exemplified by artemisinin [7], taxol [8] and ganoderic acid [9]. Some terpenoids have also been found to be involved in the regulation of growth and development [10,11,12], including cis-zeatin (CZ), trans-zeatin (TZ), abscisic acid (ABA) and gibberellin A4 (GA4). These terpenoids can be categorised into zeatins, monoterpenoids, sesquiterpenoids, diterpenoids, triterpenoids and steroids, which are synthesised by the mevalonate (MVA) pathway; and gene isopentenyl diphosphate isomerase (*IDI*), geranyl pyrophosphate synthase (GPS), farnesyl pyrophosphate synthase (FPPS), geranylgeranyl pyrophosphate synthase (GGPS), squalene synthase (SQS) and lanosterol synthase (LS), which are direct precursor genes for the biosynthesis of zeatins, monoterpenoids, sesquiterpenoids, diterpenoids, triterpenoids and steroids, respectively [13]. Despite scientific and medicinal interest in the terpenoid biosynthesis pathways of *S. baumii*, the compounds present and their biosynthesis-related genes have not been fully elucidated. Furthermore, terpenoid hormones involved in growth and development have received more attention in plants than fungi, and they may also play a distinct role in the growth and development of *S. baumii*.

Understanding the growth and development mechanism and the terpenoids present in *S. baumii* could be helpful in improving yield and quality. In the present work, we collected material from three different developmental stages of WT *S. baumii* (Myc, Pri and Fru). To clarify the growth and development mechanism, metabolomics and transcriptomics approaches were employed to analyse differences in metabolite abundance and gene expression and explore correlations between terpenoid metabolites and gene expression. The central gene, *IDI*, and four terpenoid hormones involved in growth and development were confirmed. To further explore the growth and development mechanism of *S. baumii*, we measured transcript levels of terpenoid genes and the content of terpenoid metabolites between WT and *IDI*-transformant (IT) strains. Finally, we determined that the *IDI* gene dominates the growth and development of *S. baumii* by regulating transcript levels of terpenoid biosynthesis genes and the content of terpenoid hormones. The results revealed the terpenoids present in *S. baumii* and the associated terpenoid biosynthesis genes. Additionally, terpenoid hormones related to growth and development and the biosynthesis pathway were identified. The results will likely promote the cultivation of high-quality *S. baumii* in the future.

## 2. Materials and Methods

### 2.1. Fungal Materials and Collection

WT fruiting bodies (WT Fru) were collected from *S. reticulata* at Liangshui Nature Reserve, Lesser Xing’an Mountains, Yichun city, Heilongjiang Province, China (Figure 1). Mycelia isolated from Fru were identified as *S. baumii* according to ITS sequence alignment (NCBI GenBank No. KP974834). Subsequently, mycelia were inoculated in potato dextrose (PD) medium and grown in a shaking incubator at 170 rpm and 25 °C for 10 days, and WT mycelia (WT Myc) were collected (Figure 1). Subsequently, mycelia were inoculated into polypropylene bagscontaining culture matrix to culture WT primordia (WT Pri, Figure 1). The bags were then cultured at 25 °C and 60–70% relative humidity in the dark, and WT Pri samples were collected after one month. For detailed culture methods and conditions, refer to Liu et al. [3].

### 2.2. Widely Targeted Metabolome Analysis

Three biological replicates were collected for each stage, and all samples were immediately frozen in liquid nitrogen and stored at −80 °C. Freeze-dried samples were crushed and extracted following a previously reported procedure [14]. QC samples were prepared by mixing all samples to determine the reproducibility of the obtained results. These samples were stored at −80 °C until ultrahigh-performance liquid chromatography electrospray ionisation tandem mass spectrometry (UHPLC-MS/MS) analysis(Sciex, Framingham, MA, USA). 

Quantitative analysis of metabolites was accomplished by multiple reaction monitoring (MRM). SWATHtoMRM is an R package used to construct a large-scale set of MRM transitions from the acquired SWATH-MS data files [15].SCIEX Analyst Work Station software (Version 1.6.3) was employed for MRM data acquisition and processing. MS raw data files (.wiff) were converted to TXT format using MSconventer software [15]. In-house R programs and databases were applied for peak detection and annotation [16]. Metabolome analysis was conducted by Guangzhou Magigene Biotechnology Company (Guangzhou, China). Differential metabolites (DEMs) were screened based on *p*-value <0.05 among the metabolites with VIP >1.

### 2.3. Transcriptome Sequencing and Data Analysis

For transcriptome sequencing, the sampling method was the same as that used for the metabolome analysis experiment, with three biological replicates for each sample. Total RNA was extracted using an RNAiso Plus kit (Takara, Dalian, China) according to the manufacturer’s directions. RNA samples that met the requirements were sent to Magigene Biotechnology Company (Guangzhou, China) for transcriptome sequencing on an Illumina HiSeq 2500 platform. 

Raw reads in fastq format were first analysed for quality control. Reads containing adapters, sequences with more than 10% unknown nucleotides and low-quality reads were primarily removed from the raw dataset. All subsequent analyses were based on the high-quality clean data. Nine libraries (Myc, Pri and Fru, three replicates for each treatment) were compared with the reference genome sequence of *S. baumii* (https://www.ncbi.nlm.nih.gov/genome/?term=Sanghuangporus%20baumii (accessed on 20 July 2016)), and reads were compared with the relevant reference sequence to determine sequencing saturation. Gene expression levels were calculated as fragments per kilobase per transcript per million mapped reads (FPKM), and principal component analysis (PCA) and gene expression pattern cluster mapping were performed. Genes with FDR ≤ 0.05 and |log_2_ (fold change)| ≥ 1 were designated as differentially expressed genes (DEGs), and those DEGs were annotated using KEGG pathway analysis.

### 2.4. Validation of Gene Expression from Transcriptome Data

The Illumina sequencing results were validated by qRT−PCR for 16 DEGs involved in terpenoid biosynthesis. Their primers for qRT−PCR were designed with Primer Premier 5.0 software (Appendix A). The *S. baumii* samples were the same as those used for transcriptome analysis, with three biological replicates for each treatment. qRT−PCR was performed on a CFX 96 real-time PCR detection system (Bio-Rad, Hercules, CA, USA), referring to the reaction parameters in the SYBR Green Master Mix (Takara) guide.All protocols were carried out according to the manufacturer’s instructions. Relative expressionlevels of target genes were calculated using the 2^−ΔΔCt^ method [17] with the *α*-*tubulin* gene as an internal control [3]. 

### 2.5. Adding or Spraying Exogenous Terpenoid Hormones in the Growth and Development Stage of S. baumii

A transformant strain of *S. baumii* was previously obtained, the transformation of which was mediated by *Agrobacterium* EHA105 (containing pCAMBIA1301-*gpd*-*IDI* plasmid). The visible *S. baumii* single colony was selected and transferred to fresh PDA medium containing 4 μg/mL hygromycin (Hyg). The colonies were subcultured five times to obtain a stable, positive *IDI*-transformant (IT) strain (unpublished).IT materials (IT Myc, Pri and Fru) were cultured in accordance with previous reports [3]. To test the effects of hormones CZ, TZ, ABA and GA4 (J&K Scientific, Beijing, China) on the growth and development of WT and IT *S. baumii*, 10 mg/L CZ, TZ, ABA and GA4 were separately added to PDA medium. Myc and Pri of WT and IT *S. baumii* samples with a diameter of 0.5 cm were inoculated on PDA petri dishes, and the growth rate ofMyc and the germination rate of Pri samples were observed after 10 days. WT and IT *S. baumii* Fru were sprayed with the same concentration of four exogenous hormones three times a day, and the effects of these four hormones on the dry weight of Fru samples were measured after 20 days. All experiments were repeated three times for each group.Finally, the growth rate of Myc, the germination rate of Pri and the dry weight of Fru were measured.

### 2.6. Determination of the Transcript Level of Gene and Hormone Content

Myc, Pri and Fru materials fromWT and IT *S. baumii* were collected as described for the metabolome experiment. Gene transcript levels were determined by qRT−PCR as described above. The determination method of hormone content is as follows: a 0.5 g quantity of each individual sample was weighed, crushed and mixed with 4.5 mL of phosphate-buffered saline (PBS) extract solution (pH7.4). After centrifuging at 12,000× *g* for 15 min, the supernatant was carefully transferred to a 10 mL centrifuge tube. Levels of CZ, TZ, ABA and GA4 were determined using ELISA kits (mlbio, Shanghai, China) following the manufacturer’s instructions. The total triterpenoids content was measured as described previously [3].

## 3. Results

### 3.1. Metabolome Analysis of Different Developmental Stages of S. baumii

A widely targeted metabolome analysis was performed on samples taken from different developmental stages of *S. baumii*, and 452 metabolites were obtained across all samples (Appendix A). In the PCA plot, the QC samples grouped together, indicating that they had similar metabolic profiles and that the entire analysis was reliable and repeatable. Biological replicates for each of the three different development stages clustered together in different areas, suggesting that there were significant differences in metabolites (Figure 2A). Differential metabolites wereillustrated using a Venn diagram (Figure 2B), a bar chart (Figure 2C) and volcano plots (Figure 2D). There were 51 differentially abundant metabolites (21 upregulated and 30 downregulated) between Fru and Myc. There were 32 differentially abundant metabolites (8 upregulated and 24 downregulated) between Fru and Pri. There were 28 differentially abundant metabolites (15 upregulated and 13 downregulated) between Pri and Myc. Differential metabolites include terpenoids, flavonoids, alkaloids, carbohydrates, etc. Information on their up/downregulation is provided in Figure 2D.

### 3.2. Terpenoid Analysis Related to Growth and Development of S. baumii

A total of 125 terpenoids were identified across the three different developmental stages, including 2 zeatins, 12 monoterpenoids, 23 sesquiterpenoids, 21 diterpenoids, 28 triterpenoids and 39 steroids (Figure 2E, Appendix A). Most terpenoids were found to have medicinal value, and only four terpenoid hormones related to growth and development were identified, namely CZ, TZ, ABA and GA4. ELISA results showed that CZ content was highest in the Myc stage (1.18 mg/g), TZ and GA4 levels were speculated to be highest in the Pri stage (1.27 mg/g and 843.00 pmol/g) and ABA content was highest in the Fru stage (896.11 ng/g, Appendix A). Variation in the content of the four terpenoid hormones determined by ELISA was essentially consistent with levels measured by UHPLC-MS/MS, with a correlation coefficient, R > 0.885 indicating that ELISA kit determination for all four terpenoid hormones was stable and reliable. These results tentatively suggested that CZ plays an important role in the Myc stage, whereas TZ and GA4 are important in the Pri stage and ABA is crucial in the Fru stage.

### 3.3. Differentially Expressed Genes between the Three Different Developmental Stages of S. baumii

Three independent replicates for the three different developmental stages of *S. baumii* were subjected to RNA-Seq analyses. The results showed that the three biological replicates for each of the three different development stages clustered together in different areas (Figure 3A), and the number of genes identified did not differ considerablywith a further increase above a certain threshold (Appendix A), illustrating the significant differences in genes and adequate sequencing saturation. Appendix A also shows that the three replicates for each sample clustered together, confirming the stability and repeatability of the data. Using RNA-Seq technology, 8.09 Gb, 8.59 Gb and 6.74 Gb of raw data wasobtained from samples of the three different developmental stages, and the total number of clean paired reads was 24,633,634, 26,032,769 and 20,39,8644, respectively. This resulted in approximately 6.02–7.59 Gb of clean reads, a Q20 value > 98.64%, a Q30 value > 94.94% and an average GC % ratio of 52.08% (Appendix A). The above results demonstrate that the RNA-Seq data were suitable for subsequent analyses. DEG analysis revealeda total of 4075 DEGs (1770 upregulated and 2305 downregulated) between the Fru and Myc groups. There were 3860 DEGs (1580 upregulated and 2280 downregulated) between Fru and Pri. There were 2408 DEGs (1216 upregulated and 1192 downregulated) between Pri and Myc (Figure 3B,C).

### 3.4. Genes Related to Terpenoid Biosynthesis 

All terpenoid synthesis genes were analysed using KEGG pathway classification (Figure 3D). Terpenoid backbone annotated to 21 genes, zeatin biosynthesis annotated to 1 gene, sesquiterpenoid biosynthesis annotated to 3 genes, diterpenoid biosynthesis annotated to 8 genes, triterpenoid and steroid biosynthesis annotated to 2 and 22 genes, respectively (Appendix A). The expression levels of these genes were significantly different in different stages, determining the differences in the synthesis of terpenoid products (Figure 4). Some transcripts expressed at high levels were found to encode terpenoid biosynthesis genes in *S. baumii*, including HMG-CoA synthase (HMGS), HMG-CoA reductase (HMGR), mevalonate pyrophosphate decarboxylase (MVD), *IDI*, squalene epoxidase (SE), ABA4 and ergosterol C14 reductase (ERG24, Appendix A). Some key terpenoid synthesis genes were also confirmed that were significantly correlated with the abundance patterns of their synthesised products (Figure 3D), suggesting that they likely play an active role in the biosynthesis of terpenoids.

### 3.5. Validation of Gene Expression Patterns by qRT−PCR

To validate the accuracy and repeatability of the transcriptome data, qRT−PCR was conducted on a set of DEGs associated with terpenoid metabolites. Among all candidate DEGs related to terpenoid biosynthesis, 16 genes were selected for qRT−PCR analysis. Figure 5 shows the expression levels of all 16 selected DEGs determined by qRT−PCR and RNA-Seq. Overall, the expression profiles of the 16 DEGs as determined by qRT−PCR were consistent with the corresponding FPKM values derived from RNA-Seq analysis. Most genes exhibited a similar expression pattern using both methods (R > 0.690), suggesting that the expression data obtained by RNA-Seq were reliable.

### 3.6. Effects of Four Exogenous Terpenoid Hormones on the Growth and Development of WT and IT S. baumii 

The *IDI* gene was found to be significantly correlated with the synthesis of CZ and also related to the synthesis of a common precursor of four terpenoid hormones (Figure 3D), suggesting that it may play an important role in the growth and development of *S. baumii*. Fortunately, we obtained an IT strain during the early stages of the experiment, and there were significant differences in morphological characteristics (i.e., Myc, Pri and Fru) between WT and IT *S. baumii* during culturing (Figure 6A). To test the metabolome hypothesis and explore the roles of these terpenoid hormones, we investigated the effects of adding or spraying four exogenous hormones (CZ, TZ, ABA and GA4) on the growth and development processes of WT and IT *S. baumii*.

The results showed that supplementation each of the four exogenous terpenoid hormones did not significantly affect the growth of WT Myc; rather, there was a slight decrease, possibly indicating that the hormone content in Myc had reached a balance, and there may have even been a negative feedback effect (Figure 6B,C). Thiswas also found in the WT Pri and IT Fru groups. In the IT Myc group, only the growth rate of Myc supplemented with CZ (0.32 cm/d) was faster than that of IT control Myc (0.29 cm/d, Figure 6B,C). In the WT and IT Pri groups, the addition of each of the four exogenous hormones had a significant inhibitory effect on the germination rate of WT Pri, whereas the addition of TZ (0.29 cm/d) and GA4 (0.28 cm/d) promoted the germination rate of IT Pri (Figure 6D,E). In the WT and IT Fru groups, only spraying of GA4 promoted the growth of WT Fru (0.04 g/d), whereas spraying the other three exogenous hormones had no obvious effect on the growth and development of Fru (Figure 6F,G). Upon addition of exogenous hormones, we found that CZ played a positive role in promoting the growth of *S. baumii* in the Myc stage, whereasTZ and GA4 were important in the Pri stage. These results are nearly consistent with the above metabolome results. However, GA4 had a strong influence during the Fru stage, rather than ABA as previously speculated.

### 3.7. Differences in Gene Expression and Terpenoid Content between WT and IT S. baumii 

*IDI* gene overexpression did not accelerate the growth rate during any stage, as expected (Figure 6A). To explore the causes of this phenomenon, we measured the transcript levels of terpenoid genes (Figure 7A) and the content of terpenoid metabolites (Figure 7B). By measuring the contents of the four terpenoid hormones and the transcript levels of related genes in WT and IT *S. baumii*, we found that WT Myc grew faster than IT Myc due to accumulation of CZ following increased transcription of the *TRIT1* gene. WT Pri grew faster than IT Pri, which may be due to a significant increase in the content of one or two hormones in TZ and GA4, and levels of these hormones were also regulated by correspondinggenes. IT Fru grew faster than WT Fru due to accumulation of GA4 following increased transcription of the *GGPS* gene (Figure 7), suggesting that phenotypic changes may be influenced by a single hormone or may be caused by the combined actions of two or more hormones [18].

## 4. Discussion

In this study, inan effort to elucidate the variation in terpenoid hormones and their underlying regulation in different developmental stages of *S. baumii*, a correlation analysis of metabolomes and transcriptomes was performed.

### 4.1. Application of Metabolomesin S. baumii 

Metabolomes can reflect biological changes caused by genetic variation and/or environmental disturbances [19]. Through metabolome analysis, we found a total of 125 terpenoid metabolites of *S. baumii* in different developmental stages. Compared with NMR technology [20], the number of terpenoids identified by the metabolome was significantly increased. Obviously, UHPLC-MS/MS technology is more advantageous in fungi with complex terpenoid metabolites. Among the 125 identified terpenoid metabolites, most have good medicinal value, including anti-inflammatory [21], antianxiety [22], anticancer [23,24,25], antioxidation [26] and cholesterol-lowering [27] activities, and the detection of these terpenoids in *S. baumii* also confirmed the extremely high medicinal value of *S. baumii* [1,2,3]. Additionally, we found four terpenoid hormones related to growth and development in *S. baumii* that play a key role in different developmental stages.

### 4.2. Application of Transcriptomesin S. baumii

Transcriptomescan provide full coverage of transcript abundance expression [28]. We excavated 57 genes involved in terpenoid synthesis and some high-transcript genes based on transcriptomes [13,29], which were annotated to the MVA pathway (Appendix A). The results confirmed that fungal terpenoids are synthesised through the MVA pathway rather than the MEP pathway [29,30]. Furthermore, we found that TZ synthesis genes and ABA synthesis genes were not annotated in zeatin and sesquiterpenoid biosynthesis pathways. Based on previous reports, we obtained *CYP5340A71*, *CYP5340A72* (named by the P450 naming committee) and *ABA4* genes based on homology. The *CYP5340A71* and *CYP5340A72* genes of *S. baumii* clustered together with genes *AtCYP735A1* and *AtCYP735A2* of *Arabidopsis* [31]. The *ABA4* gene of *S. baumii* clustered with *Botrytis cinereaBcABA4* (Appendix A) [32]. The results prove that the terpenoid biosynthesis genes used in this study arereliable and also considerably enriched the terpenoid biosynthesis gene database of *S. baumii*.

### 4.3. Correlation of Terpenoid Metabolites and Relative Genes of S. baumii

Metabolome and transcriptome analyses have become common technical means to explore the underlying molecular mechanisms of fungal metabolic biosynthesis [28,30], and their combination can provide an in-depth and comprehensive understanding of the relationship between gene function and metabolites and/or phenotypes [5,33]. In this study, we observed a correlation and linkage between terpenoid products and terpenoid synthase gene expression levels at different developmental stages (Figure 3D and Figure 5). For instance, the highest CZ product abundance and *IDI* gene expression level were observed in Myc, followed by Pri and Fru. Furthermore, we revealed a correlation and linkage between TZ and *CYP5340A71* and *CYP5340A72* gene expression levels in different developmental stages. Similarly, the highest ABA product abundance and the highest *ABA4* gene expression levels were observed in Fru, followed by Priand Myc. Our results are in agreement with those of previous studies [29,30], showing that terpenoid products are mainly controlled transcriptionally. The function of these genes remains to be further investigated.

### 4.4. Exploration of Growth and Development Mechanisms of S. baumii

Analysis of the underlying mechanism revealed the regulation of growth and development of *S. baumii*. We compared the differences in terpenoid hormone contents and related gene transcript levels between IT and WT *S. baumii* and found that overexpression ofthe *IDI* gene does not increase the contents of all terpenoid hormones, mainly due to competition among hormones and tissue-specific expression [34,35]. IT *S. baumii* CZ, TZ and GA4 levels and transcription of related genes were decreased in Myc and Pri stages, presumably due to increased triterpenoids accumulation caused by high-level expression of triterpenoid biosynthetic genes (Figure 7). When the copy number of triterpenoid biosynthesis genes was low, the triterpenoid content was significantly decreased, and the GA4 content and transcription levels of genes related to GA4 biosynthesis were significantly increased; hence, IT Fru grew faster than WT Fru (Figure 7). This also explains why GA4 had a strong influence during the Fru stage rather than ABA. ABA seems to be an indicator of maturity, and its content is only higher in fast-growing or large tissues (Figure 6A and Figure 7B). This phenomenon was also observed in the growth and development of strawberry [36], suggesting that ABA is related to maturation; however, whether maturation depends entirely on the ABA4 gene expression remains to be further verified.

Studies have shown that the growth and development of *S. baumii* are affected by terpenoid hormones, althoughit is unknown whether expressing all terpenoid hormone synthesis genes at the same time can accelerate the whole growth and development stage. Therefore, in future studies, we plan to use the gene coexpression method to study the growth and development of *S. baumii* [9] in an attempt to break the phenomenon of competition among hormones and tissue-specific expression [34,35]. The results will be further explored to provide a better theoretical basis for the breeding of excellent varieties of *S. baumii*.

## 5. Conclusions

Metabolomics and transcriptomics approaches were applied to identify four terpenoid hormones and the associated key biosynthesis gene, *IDI*. Subsequently, we measured the transcript levels of terpenoid genes and the content of terpenoid hormones. Analysis of the underlying mechanism revealed that the *IDI* gene repressed the transcription of *TRIT1*, *CYP5340A71*, *CYP5340A72* and *GGPS* genes by increasing the transcript levels of *SQS* and *LS* genes, and this consequently decreased the accumulation of CZ, TZ and GA4, which slowed growth in the Myc and Pri stages. The *IDI* gene increased transcription of the *GGPS* gene by attenuating the transcriptional activity of *SQS* and *LS* genes, resulting in increased GA4 content and accelerated growth in the Fru stage. Specifically, CZ played a major role in the Myc stage, TZ and GA4 functioned in the Pri stage and GA4 influenced the Fru stage. ABA may be a marker of maturity in *S. baumii*. These findings provide important guidance for molecular breeding of *S. baumii*.

## Figures and Tables

**Figure 1 jof-08-00648-f001:**
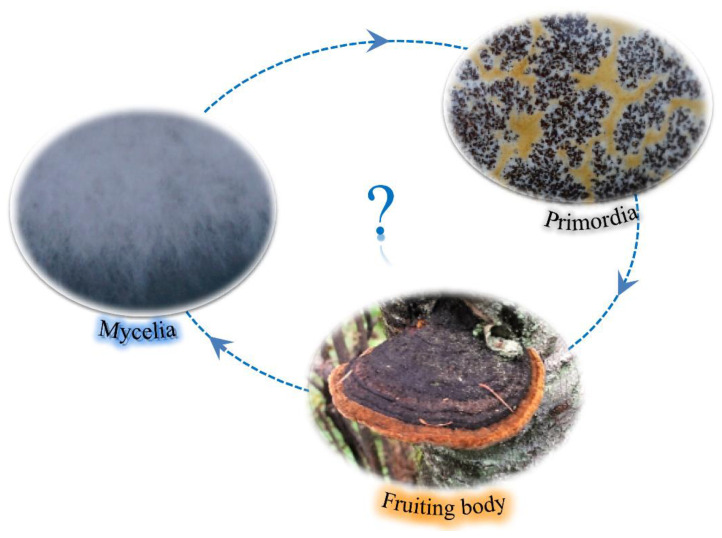
The growth and development process of *S. baumii*. “?” indicates an unclear growth and development mechanism.

**Figure 2 jof-08-00648-f002:**
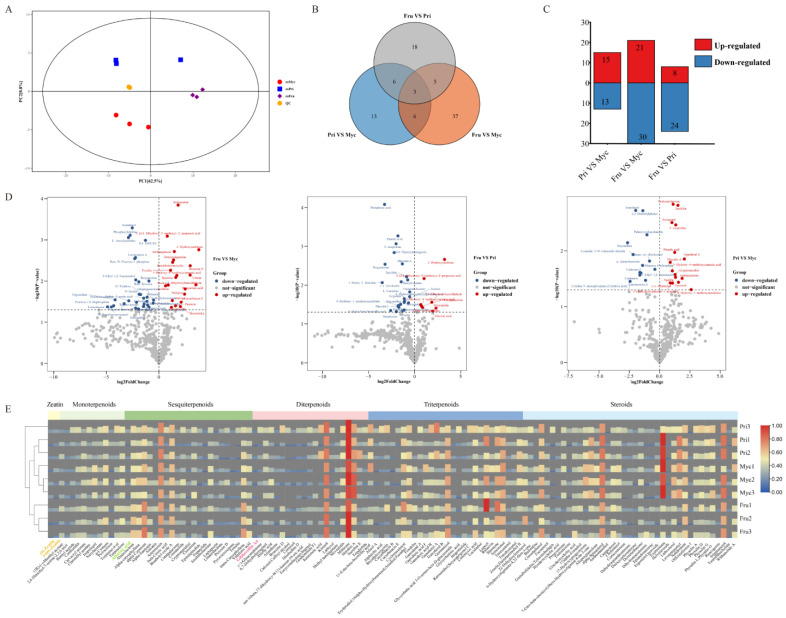
Metabolome analyses of different developmental stages of S. baumii. (**A**) PCA score plot of metabolite profiles of Myc, Pri and Fru stages of *S. baumii*. (**B**) Venn diagramshowing up- and downregulated DEMs based onpairwise comparisons of Fru vs. Myc, Fru vs. Pri and Pri vs. Myc stages. (**C**) Bar graph of up- and downregulated DEMs based onpairwise comparisons. (**D**) Volcano map of up- and downregulated DEMs based onpairwise comparisons. (**E**) Hierarchical clustering of terpenoid DEMs in Myc, Pri and Fru stages. Rectangles with different colours represent different terpenoids.

**Figure 3 jof-08-00648-f003:**
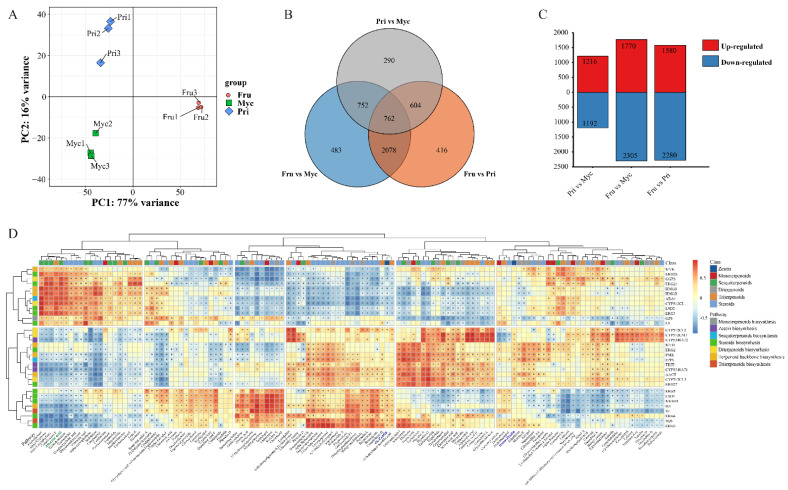
Transcriptome analyses of different developmental stages of *S. baumii*. (**A**) PCA score plot of transcript profiles from Myc, Pri and Fru stages. (**B**) Venn diagram of up- and downregulated DEGs based on pairwise comparisons of Fru vs. Myc, Fru vs. Pri and Pri vs. Myc stages. (**C**) Bar graph of up- and downregulated DEGs from pairwise comparisons. (**D**) Correlation of DEGs and DEMs involved in terpenoid biosynthesis in *S. baumii*. Square patterns with different colours represent different terpenoid biosynthesis pathways and class.

**Figure 4 jof-08-00648-f004:**
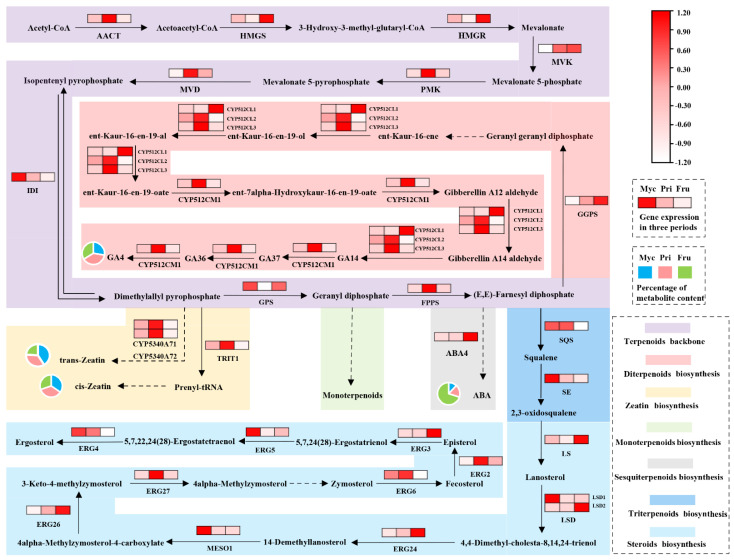
Pathways and genes involved in the biosynthesis of terpenoids during the different developmental stages of *S. baumii*. The differential expression of each annotated gene is presented as a heatmap on the corresponding place of the gene with the scale ranging from white (low) to red (high).Square patterns with different colours represent the percentage of metabolite content in comparisons of different developmental stages. The pie chart patterns with different colours represent terpenoid hormone levels at different stages of development. Rectangles with different colours represent different terpenoid biosynthesis pathways. Dotted arrows represent process unknown; solid arrows represent process known.

**Figure 5 jof-08-00648-f005:**
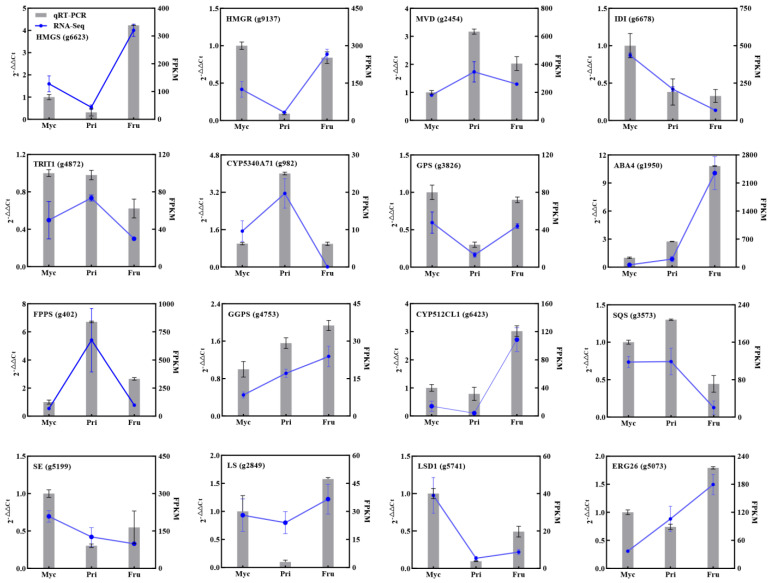
qRT−PCR validation of the expression levels of genes associated with terpenoid metabolites at the three developmental stages of *S. baumii*. qRT−PCR expression levels were calculated as a ratio relative to the level of expression in the Myc stage, which was set as 1.

**Figure 6 jof-08-00648-f006:**
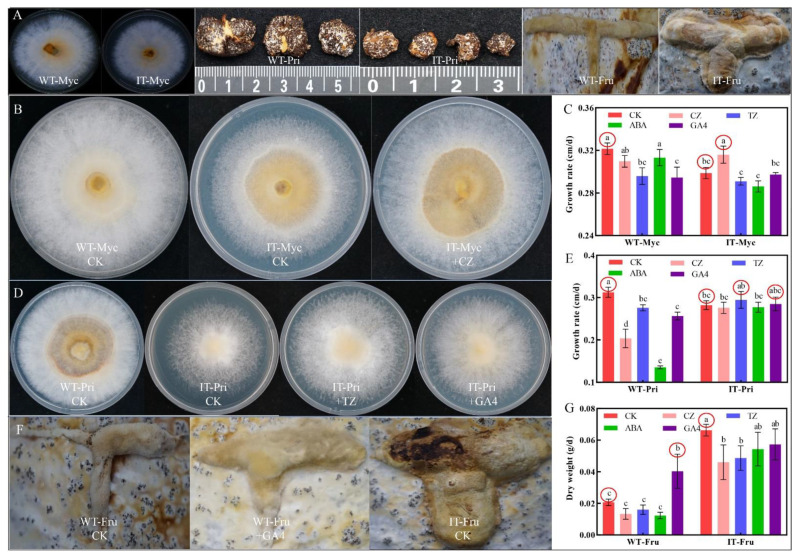
Effects of four hormones on the growth and development of *S. baumii*. CK is control. *p* < 0.05. (**A**) Differences in morphological characteristics between WT and IT *S. baumii* in the three developmental stages. (**B**) Effect of adding CZ on IT Myc growth. (**C**) Effects of four terpenoid hormones on growth rates of WT and IT Myc. (**D**) Effects of TZ and GA4 on the growth of germinating mycelia in IT Pri. (**E**) Effects of four terpenoid hormones on the growth rate of germinating mycelia in WT and IT Pri. (**F**) Effect of GA4 spraying on the growth of WT Fru. (**G**) Effects of spraying four terpenoid hormones on dry weight of WT and IT Fru.

**Figure 7 jof-08-00648-f007:**
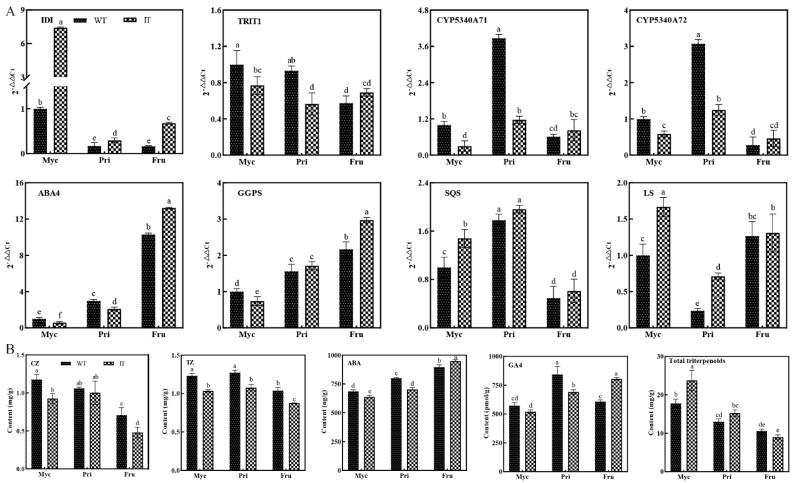
Differences interpenoid metabolite content and gene transcript levels between WT and IT *S. baumii* observed in the three developmental stages. *p* < 0.05. (**A**) Differences in transcript levels of terpenoid biosynthetic genes between WT and IT *S. baumii* in the three developmental stages. (**B**) Determination of terpenoid content differences between WT and IT *S. baumii* in the three developmental stages by ELISA.

## Data Availability

Not applicable. All data are contained within the article.

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
