# Peer review of "Metabolome and Transcriptome Profiling Reveal That Four Terpenoid Hormones Dominate the Growth and Development of *Sanghuangporus* *baumii"

_jof, 2022, doi:10.3390/jof8070648_

Round 1

Reviewer 1 Report

This is a very interesting manuscript about fungal terpenoids as a regulators of growth and development. This study uses the combined analysis of transcriptome and metabolome to reveal terpenoids up- or down-regulated on three developmental stages. The importance of the four terpenoids was confirmed by addition of exogenous hormones to fungal colony. The study was carried out very carefully, at a high professional level. A large amount of new information about the role of terpenoids in fungi was obtained.

Although the manuscript is carefully written and the presented experiments seem to be well conducted, however, there are few questions to the authors.

(see the file)

Author Response

Thanks for the comments. All comments are constructive, and revisions have been made based on the suggestions.

  1. What equipment was used for UHPLC-MS/MS analysis? (2.2)

A: Sciex equipment was used for UHPLC-MS/MS analysis.

  1. Please clarify in 2.4 what method was used for the qRT-PCR analysis (SYBR chemistry)?

A: SYBR chemistry was used for the qRT-PCR analysis, it has been added to the manuscript.

  1. Please clarify in 2.5 the manufacturer of hormones CZ, TZ, ABA and GA4.

A: Hormones CZ, TZ, ABA and GA4 were purchased from J&K Scientific (Beijing, China), which have been added to the manuscript.

  1. Algorithms of metabolite identification and quantification require a more detailed description in 2.2. The list of specific MRM transitions that allow to determine terpenoid metabolites during MRM-profiling would be very helpful.

A: Algorithms of metabolite identification and quantification were performed in Ref. [15]. Some descriptions and Ref. have been added to the manuscript.

[15]Zha, H.H.; Cai Y.P.; Yin, Y.D.; Wang Z.Z.; Li, K.; Zhu Z.J. SWATH to MRM: Development of high-coverage targeted metabolomics method using SWATH technology for biomarker discovery. Anal. Chem. 2018, 90, 4062–4070. DOI: 10.1021/acs.analchem.7b05318

SWATHtoMRM workflow. SWATHtoMRM is an R package to construct a large-scale set of MRM transitions from the acquired SWATH-MS data files, which is freely available on the Internet (http://www.zhulab.cn/software.php). It mainly includes two parts: untargeted analysis of SWATH-MS data and generation of MRM transitions.

(1) Untargeted analysis of SWATH-MS data. The raw SWATH-MS data (.wiff files) were first

converted to mzXML files using the "msconvert” program form ProteoWizard (version 3.0.6428).

Multiple data files were grouped in one folder and processed by SWATHtoMRM. There are 4 steps for the analysis of SWATH-MS data: (1) MS1 peak detection and alignment; (2) extraction of MS2 peaks and chromatograms; (3) MS1 & MS2 peak grouping; (4) generation of consensus MS2 spectrum.

(2) Generation of MRM transitions. Each product ion in the consensus spectrum of a specific

metabolite was further evaluated to generate the MRM transitions. There are three criteria for the

selection of MRM transitions: 1) m/z(product ion ) < m/z(precursor ion)−14.0126 Da, which means that at least a “CH2” group (14.0156 Da ± 3 mDa) is lost from the precursor ion upon fragmentation; 2) eliminating the product ions with the loss of H2O, NH3 or CO2, and users can include different neutral losses in the program; 3) choosing the product ion with the highest intensity among the remaining product ions. Here, each metabolite has one chosen MRM transition. Finally, a csv file containing all generated MRM transitions was output for targeted analysis.

  1. Levels of CZ, TZ, ABA and GA4 were determined using ELISA kits (mlbio, Shanghai, China) following the manufacturer’s instructions. As far as I know, ELISA kits are designed to determine hormones in blood plasma. Have you had a preliminary testing of the method to guarantee that these kits can be applied to fungi?

A: We have confirmed with manufacturer that ELISA kits are suitable for fungal hormone assays, and the specific procedures have been added to the supplementary data.

  1. Figure S4. Phylogenetic analysis of terpenoid biosynthesis genes of S. baumii.

The figure requires explanation in the legend. Is it a phylogenetic comparison of S. baumii genes and genes of other fungi or not only fungi? Fm, Pn, Sp, Ps, Rv – these are the first letters of organisms' names, I suppose? What organism do they correspond? It would be nice to write Genbank accession numbers of all these genes in the legend. The phylogenetic tree was generated by…? 

A: The phylogenetic tree was generated by MEGA. Genbank accession numbers of all these genes have been added to supplementary data.

  1. Figure 6. Add to legend, that "CK" is control. And what is letters a, ab, abc over the bars.

A:"CK is control" is added to the legend. The letters a, ab, abc indicate significant differences, p < 0.05

  1. In WT and IT Pri groups, addition of each of the four exogenous hormones had a significant inhibitory effect on germination rate of WT Pri, while addition of TZ (0.29 cm/d) and GA4 (0.28 cm/d) promoted the germination rate of IT Pri (Figure 6D, E). … Upon adding exogenous hormones, we found that CZ played a positive role in promoting the growth of S. baumii in the Myc stage, while TZ and GA4 were important in the Pri stage. These results are consistent with the above metabolome results.

Did you really estimate the rate of germination of fungal spores or do you mean growth rate of colonies? If you mean growth rate, is statistically significant the difference between control IT-pri and IT-pri+TZ and between control IT-pri and IT-pri+GA4? I do not agree with the conclusion that the level of TZ was highest in the primordium stage. According to the Figure S1, we only can conclude that the level of TZ in fru stage is lower than in myc and pri stages (ELISA results). Measurements by UHPLC-MS/MS do not allow a statistically reliable conclusion due to a large variability.

A: It means growth rate, we have revised this section as you suggested. “TZ and GA4 level were speculated to be highest in the Pri stage (UHPLC-MS/MS results)”

  1. Figure 7. Please clarify in the legend that B demonstrates the level of hormones measured using ELISA (if it is true).

A: The level of hormones was determined by ELISA, which has been added to the manuscript.

  1. In my opinion, there is not enough discussion (comparison with literature) of the fact that terpenoid compounds can play the role of development regulators, hormones, in fungi. All references about the role of terpenoids in the development that are cited in the manuscript concern plants. And what about fungi ? Are there any reports about terpenoid hormone biosynthesis in fungi? I managed to find only one job about fungi - plant pathogens https://doi.org/10.1111/mpp.12393 In addition to the production of  canonical effectors, fungi also produce compounds that are similar to plant hormones, such as auxins, cytokinins (CKs), gibberellic acids (GAs), ethylene (ET), abscisic acid (ABA), jasmonic acid (JA) and salicylic acid (SA).

A: Terpenoid hormones are mostly studied in plants, but hardly reported in fungi. We also noticed that terpenoid hormones can affect growth and development by overexpressing the terpenoid synthesis gene IDI, so we did not discuss the role of terpenoid hormones in fungi. In the future, we will further study the mechanism of action of these terpenoid hormones in fungi.

Minor comments

  1. Introduction. Apparently, there is an extra word in the sentence: "The central gene IDI and four terpenoid hormones involved in growth and development were identified confirmed."

A: “identified” has been removed

  1. In order to further study the underlying regulation of growth and development of S. baumii. (4.4 ) - rephrase please.

A: This sentence has been revised in the manuscript

Reviewer 2 Report

The manuscript addresses an interesting subject but fails to provide a complete exercise of the scientific methodology.
Metabolomic and transcriptomic approaches, by themselves, are no longer considered tools that provide robust results that can stand alone in a publication. Instead, they provide observations that help elaborate hypotheses that experimentation will accept or reject. The authors identified the IDI gene as a putative regulator of S. baumii growth and development, and this central hypothesis should be tested by molecular means. The authors mention that an IDI transformant was generated, but no details about this are provided. Is it a null mutant? Silencing mutant? Edited mutant? Details about the generation and molecular characterization should be included in the manuscript. In addition, control strains to associate the phenotype with genetic modification should be also included.
Regarding the validation by RT-qPCR, please include references that support that the alpha-tubulin encoding gene can be used as a control for data normalization in S. baumii expression assays.

Author Response

Thanks for the comments. All comments are constructive, and revisions have been made based on the suggestions.

  1. We used transgenic methods to test the IDI gene. IDI transformant is an edited mutant, which is an IDI gene overexpressing strain (unpublished). Morphological characteristics of the IDI transformant and control strains are provided in Figure 6A, and some details have been re-described in the manuscript.

  1. The alpha-tubulin encoding gene was verified to be a better control in previous studies, and the Ref. [3] has been added to the manuscript.

Reviewer 3 Report

The manuscript in reference describes a comparative study on the metabolome and transcriptome variations, depending on developmental stages, of Sanghuangporus baumii. The main strength is the aim of the study regarding the combined variations of metabolites and genes for the first time. Methods seem to be well-conducted and adequately described, although a few details are missing (vide infra). Data seem to be well-interpreted. In my opinion, the manuscript is interesting and has important results. However, some issues should be addressed prior to further consideration.

1.       Detailed scrutiny throughout the manuscript to revise some grammar and stylistic issues is recommended since some passages and sentences are difficult to be followed.

2.       Title: Terpenoids are a vast group of metabolites, and it is well-known that triterpenoids in agaricomycetes have several functions. Therefore, I recommend that the authors specify which kind of terpenoids (and even which type of triterpenoids) are involved in the growth and development of S. baumii. This suggestion should be consistent in the manuscript.

3.       Abstract order is incorrect and can be re-organized since some results are provided before the aims and procedures.

4.       Abstract (ending) the word “illuminate” is not correct.

5.       M&M section needs to be carefully revised. Some experimental details are missing to ensure outcome reproducibility. For instance, brand, model, and grade of reagents, solvents, materials, and instruments must be provided. In addition, details of how levels of CZ, TZ, ABA, and GA4 were determined using ELISA kits should be provided.

6.       Page 2: Specify the ITS sequence alignment to identify the mycelia as S. baumii.

7.       Page 3: more details about the growth stage of  S. baumii must be provided.

8.       Section 3.1.: Specify which metabolites (not only the number) were up/down-regulated.

9.       Figure 2. This figure must be enlarged since it is challenging to see the information contained in this figure.

10.   Section 3.2.: was really hormones determined by ELISA? (I mean, through an enzyme immunoanalysis?) Or through a colorimetric test? More details must be provided in M&M section as suggested (vide supra).

11.   The discussion section can be improved since it is highly descriptive and even speculative. Accordingly, apart from expanding the descriptions of obtained results, the discussion should be oriented towards discussing the results from a mechanistic point of view and the plausible relationships supported by previous studies, theory, and literature.

12.   Conclusions section can also be improved since it is a summary of results, and no conceptual findings are provided.

13.   Finally section 2.2: the manuscript needs to improve the information related to the confidence levels of the communicated identity of the compounds in Table S2. Triterpenes are specialized metabolites with rich isomerism, such as regio and stereoisomerism, which is very difficult to accurately define even by HRMS and MS/MS. Using standards could overcome such an issue if co-chromatography indicates the presence, although it is not a simple procedure if co-elutions with very similar compounds are presented. In this sense, the manuscript has a problem related to the informed identity of the 125 terpene-related metabolites provided in Table S2 and Figure 2. In fact, a clear explanation about such an identification process is not provided in M&M nor discussed. Although the tentative identification is valid, I consider that authors did not adequately communicate the confidence levels of these identities, since I am not convinced that triterpenes with identical accurate mass (AM) can be discriminated (even including stereoisomerism) by HRMS and some soft ionization-derived fragments. For instance, compounds such as bornyl acetate and neryl acetate have identical molecular formula (MF) and AM (i.e., MF = C12H20O2, AM = 196.14XX). Similar condition for compounds shionone, simiarenol, taraxasterol, and citrostadienol, and others in the list (MF = C30H50O, AM = 426.38XX). Is the identity of these compounds adequately informed? Can the authors assure that there is no exchanged identity?

14.   In addition, based on the previous comment, I can't entirely agree as well with the classification according to the compounds class (column 6 in Table S2) since the accurate identification is not ensured nor confidently communicated, so the information derived from such a partial or tentative identification is not adequately scoped. A classification like this and the subsequent explanation and discussion should only be performed from the accurate identification (level 1) using standards.

Therefore, in order to ensure appropriate communication of results, the adequate scope of the manuscript, and avoid confusing interpretations by readers, I cordially recommend to authors following two actions to improve the scientific quality and merit of this manuscript related to the informed metabolite identity: 

a.       Remove the proposed classification regarding the compound class and the explanations and discussion derived from that since a classification like this has no sense if accurate and unambiguous identification is not ensured. These results are overestimated with the analytical method and the interpretation performed, which could confuse readers. The comparative study looking for differential metabolites is very interesting and could be enough to be informed in the manuscript, in addition to the transcriptomic and gene expression analysis. Such an insufficiently-supported classification can be considered as a noise of this study. Be consistent throughout the manuscript.

b.      Improve the identity communication of detected metabolites listed in Table S2. In this regard, I recommend that authors use the confidence levels to communicate compound identity by HRMS. Authors can follow the recommendations of this paper (10.1021/es5002105) to appropriately communicate the identification confidence level of the detected metabolites by HRMS. In this regard, an additional column in Table S2 with such confidence levels per metabolite can be adequately added since some metabolites were identified with standards, others exhibited good MS-derived fragmentations, and other features have information very limited to the accurate mass. In this sense, avoid including overestimated identities. If previous papers reported such identification manner, it is part of a recent problem, and it is required to improve such an identity communication of metabolites analyzed by HRMS, according to the Metabolomics Standard Initiative (MSI). This condition worsens if the target molecules, such as terpenes, are structurally complex and diverse.

Author Response

Thanks for the comments. All comments are constructive, and revisions have been made based on the suggestions

A1: We have reviewed the entire manuscript in detail and revised some grammar and stylistic issues.

A2: There are mainly four terpenoid hormones related to growth and development, so I changed the title to "Metabolome and transcriptome profiling reveals that four terpenoid hormones dominate the growth and development of Sanghuangporus baumii", do you think it is possible?

A3: Abstract order has been re-organized.

A4: “illuminate” was replaced with “reveal” in the manuscript.

A5: Some experimental details have been added to the manuscript, including brand, model, and grade of reagents, solvents, etc. In addition, details of the determination of CZ, TZ, ABA, and GA4 levels using ELISA kits were provided in Supplementary Data.

A6: Mycelia isolated from Fru were identified as S. baumii according to ITS sequence alignment, and the ITS sequencing results were submitted to the NCBI database (NCBI GenBank No. KP974834).

A7: Some details about the growth stage of S. baumi have been provided in the manuscript, including temperature, humidity, incubation time, etc.

A8: Differential metabolites include terpenoids, flavonoids, alkaloids, carbohydrates, etc. Their up/down-regulated information is provided in the manuscript.

A9: Figure 2 has been enlarged, and we have uploaded high quality images in the system.

A10: Section 3.2 was determined by a colorimetric test, its principle is ELISA. Details of the assay have been provided in the M&M section.

A11: Some highly descriptive has been added to the discussion section.

A12: We have made some changes to the conclusions section.

A13: The identification of these compounds is indeed a preliminary identification, and compounds with the same molecular formula are difficult to distinguish. If some compounds need to be studied in the future, we will take your suggestions for experiments.

A14: We fully agree with your suggestion and have removed the classification according to the compounds class (column 6 in Table S2). Our experiment uses widely targeted metabolome analysis, and cannot get the confidence level similar to those obtained with HRMS. So we added the base peak chromatogram of metabolites in supplementary data.

Reviewer 4 Report

In my opinion, the submitted manuscript for review is of a high scientific standard. The various chapters of this manuscript are consistent and well written. I believe that the work should be accepted for printing after a slight correction of some Figs. Namely, Figs. 2, 3, 4, 5 and 7 are difficult to read, therefore their quality should be improved.

Author Response

Thanks very much for the suggestion. We have uploaded high quality images in the system.

Round 2

Reviewer 2 Report

I revised this second version of the manuscript, and I regret to mention that my previous recommendation stands. None of the main concerns has been properly addressed. Thanks for informing that the strain is an overexpressing mutant, but details about its generation and proper controls must be disclosed and included in the manuscript. Regarding the second point, the authors included a reference that does not validate that gene as a proper control for data normalization in RT-qPCR. Evidence that Ct values are constant in all the strains and conditions tested should be included in this study.

Author Response

Thanks for the comments. We provide some evidence and modifications through your suggestions. If you are not satisfied, please contact us

  1. The process of obtaining IDI transformants. Some descriptions were added in 2.5.

The IDI gene was amplified from cDNA of S. baumii. The amplified fragment was ligated into the vector pCAMBIA1301-gpd (linearized with the NcoI restriction enzyme) to create the pCAMBIA1301-gpd-IDI plasmid by using an In-Fusion HD Kit (Takara, Dalian, China). The pCAMBIA1301-gpd-IDI plasmid was transformed into A. tumefaciens and pre-induced for 10 min using IM medium. A suspension of S. baumii WT mycelia were treated twice by dounce homogenizer, 0.2 g S. baumii broken mycelia were mixed with 1 mL pre-induced A. tumefaciens EHA105 solution for 20 min. A 200 μL portion of the mixture was poured onto 15 mL of Co-IM medium. After incubation at 25°C, 5 mL of SM medium was covered on the mixture. After 15 d of incubation at 25°C, the visible S. baumii single colony was selected and transferred to fresh PDA medium containing 4 μg/mL hygromycin (Hyg). The colonies were subcultured five times to obtain stable IDI transformants.

  1. The alpha-tubulin encoding gene was verified to be a better control in these studies [1-2], and the transcript levels of α-tubulin gene at different developmental stages were provided in table1, it is stable.

[1]de Vega-Bartol José J,Santos Raquen Raissa,Simões Marta,Miguel Célia M. Normalizing gene expression by quantitative PCR during somatic embryogenesis in two representative conifer species: Pinus pinaster and Picea abies.[J]. Plant cell reports,2013,32,  715-729 . DOI: 10.1007/s00299-013-1407-4

[2]Bai-zhong ZHANG,Jun-jie LIU,Guo-hui YUAN,Xi-ling CHEN,Xi-wu GAO. Selection and evaluation of potential reference genes for gene expression analysis in greenbug (Schizaphis graminum Rondani)[J]. Journal of Integrative Agriculture,2018,17: 2054-2065.

DOI: 10.1016/S2095-3119(18)61903-3

Table1. Transcript levels of α-tubulin gene at different developmental stages in this study

α-tubulin

Test 1

Test 2

Test 3

Myc

23.25938

22.89063

22.89063

Pri

22.20854

22.28496

22.28496

Fru

22.23109

21.75649

21.75649